# The Generation of a H9N2 Avian Influenza Virus with HA and C3d-P29 Protein Fusions and Vaccine Development Applications

**DOI:** 10.3390/vaccines13020099

**Published:** 2025-01-21

**Authors:** Xue Pan, Fan Zhou, Xiaona Shi, Qinfang Liu, Dawei Yan, Qiaoyang Teng, Chunxiu Yuan, Bangfeng Xu, Zhifei Zhang, Minghao Yan, Zejun Li

**Affiliations:** Shanghai Veterinary Research Institute, Chinese Academy of Agriculture Sciences, Shanghai 200241, China; panxue@shvri.ac.cn (X.P.); 17601381224@163.com (F.Z.); shixiaonare@163.com (X.S.); liuqinfang@shvri.ac.cn (Q.L.); yandawei@shvri.ac.cn (D.Y.); tengqy@shvri.ac.cn (Q.T.); yuanchx@shvri.ac.cn (C.Y.); xubangfeng@shvri.ac.cn (B.X.); nzhangzhifei@163.com (Z.Z.); m17519478956@163.com (M.Y.)

**Keywords:** maternal-derived antibody, vaccine, H9N2 avian influenza virus, hemagglutinins, complement

## Abstract

Background: Maternal-derived antibody (MDA) interferes with immune responses, leading to the failure of H9N2 avian influenza vaccinations in poultry. So far, none of the commercially available H9N2 avian influenza vaccines used in poultry have been able to overcome MDA interference. Methods: To develop a vaccine that can overcome MDA interference, one or multiple copies of the minimum-binding domain (P29) from the complement protein C3d were inserted in between the signal peptide and the head domain of the hemagglutinin (HA) protein on a H9N2 avian influenza virus (A/Chicken/Shanghai/H514/2017, named H514). Results: The HA proteins containing P29 stimulated stronger type I interferences than wild-type HA proteins in vitro. The modified viruses with the HA proteins containing one copy of P29 (rH514-P29.1) and two copies of P29.2 (rH514-P29.2) were successfully rescued using reverse genetics. The inactivated vaccines developed with rH514-P29.1 or rH514-P29.2 induced higher and faster humoral immune responses than the vaccine developed with rH514 in specific pathogen-free (SPF) chickens. To evaluate the vaccines’ efficacy in the presence of MDA and to ensure a uniform level of MDA, passively transferred antibody (PTA) was used as a model to mimic MDA in 1-day-old SPF chickens. Our results showed that the rH514-P29.2 inactivated vaccine induced significantly higher HI titers than the rH514 inactivated vaccine in the presence of PTA. More importantly, it reduced viral shedding after being challenged with H514 in the presence of PTA. Conclusions: Our results suggest that vaccine antigens fused with two copies of P29 can decrease the interference of MDA on immunity in chickens. Overall, our results provide a new strategy for overcoming MDA interference.

## 1. Introduction

The H9N2 avian influenza virus (AIV) is the most harmful and widespread low pathogenicity avian influenza virus (LPAIV) in the world. The H9N2 AIV donates partial or even whole sets of internal genes to other AIVs, such as the highly pathogenic avian influenza viruses H5N1, H5N6, and H7N9, posing a substantial threat to public health [1,2,3]. In addition, co-infection of H9N2 AIV with pathogens such as infectious bronchitis virus (IBV), *Mycoplasma gallisepticum*, *Staphylococcus aureus*, *Escherichia coli*, and/or immune suppressing agents enhance the severity of the chickens’ clinical syndrome and leads to higher rates of mortality in poultry, causing enormous economic losses [4,5,6,7]. Inactivated vaccines are primarily used to prevent H9N2 AIV and have been used in poultry in China for nearly 30 years [8]. Although the vaccines are effective in specific pathogen-free (SPF) chickens in laboratories, they are less likely to be as effective in poultry, leading to constant H9N2 AIV transmission in the field [9,10,11].

Maternal-derived antibody (MDA) is one of the main factors accounting for the failure of H9N2 AIV vaccination in the field [12]. MDA is derived from maternal placental blood circulation and milk in humans and mammals, or from the yolks of eggs in avian species. On one hand, MDA can protect offsprings from many infectious diseases at the beginning of their lives when they are vulnerable [13,14,15]. On the other hand, MDA hinders immune responses to vaccinations against AIVs [12,16], Newcastle disease virus (NDV) [17,18], and the infectious bursa disease virus (IBDV) [19] in avian species, and against the measles virus in humans [20,21]. MDA decreases with age, but its interference with vaccine-based immunization leads to a transient window of increased vulnerability to infections in young animals and infants. Given the convenience and cost-effectiveness of infant vaccination, it is crucial to develop new vaccines for young animals to overcome MDA interference and elicit strong immune responses.

The complement system plays an important role in immune responses. The complement C3d covalently attaches to the microbial antigen, leading to markedly enhanced adaptive immune responses against that antigen [22,23,24]. Several studies have shown that antigens that attach to different copies of C3d (ranging from one to six) can enhance immunogenicity and induce higher levels of both humoral and cellular immune responses compared to antigens without C3d [25,26,27,28,29]. P28 (known as P29 in avian species), which is the minimum-binding domain of C3d, has also been shown to greatly enhance antibody responses and Th2 immune responses when attached to antigens [25,30,31,32]. Recently, the booster dose of a part P28-associated vaccine was reported to have overcome MDA interference in pigs [33]. However, it remains unclear whether P29 or multiple copies of P29 can overcome MDA interference in avian species.

MDA–antigen complexes are more likely to bind to B cell receptors (BCRs) and Fcγ-receptor IIB (FcγRIIB), which negatively regulate B cells activation and thus interfere with humoral immune responses [34,35,36]. Type I IFN is reported to abolish this negative regulation and promote B cells activation, even in the presence of MDA in cotton rats. This is because Type I IFN has two receptors: the traditional receptor IFNA-R and the newly found receptor CD21 (CR2), both of which are highly expressed on the surface of B cells [34]. Interestingly, CD21 is also the receptor for C3d. While binding to C3d, CD21 generates a downstream signal through CD19 immunoreceptor tyrosine-based activation motifs to activate B cells [37]. Therefore, we hypothesis that C3d and its minimum-binding domain P29 may be capable of promoting B cell activation, even in the presence of MDA in chickens.

To test our hypothesis, we inserted different copies of P29 between the signal peptide and the head domain of a H9N2 AIV hemagglutinin (HA) protein, and successfully rescued two modified H9N2 AIVs by using reverse genetics. The efficacy of the inactivated vaccines made with the two modified H9N2 AIVs were evaluated based on their ability to overcome MDA interference in chickens. MDA is naturally inherited from dams and have a high degree of variability in individual broilers, which makes it difficult to conduct MDA-related research [38]. Hyperimmune serum, which contains mostly IgY, has similar isotype proportions to MDA and can be used to mimic the presence of MDA in SPF chickens [39,40,41]. Therefore, to ensure a uniform level of MDA, passively transferred antibody (PTA) as a model to mimic MDA in 1-day-old specific pathogen-free (SPF) chickens were used.

## 2. Materials and Methods

### 2.1. The Animals and Viruses

SPF chicken eggs were purchased from the Beijing Merial Vital Laboratory of Animal Technology and hatched in the Etiologic Ecology of Animal Influenza and Avian Emerging Viral Disease (SHVRI) laboratory. All of the chickens were tagged and housed in high containment chicken isolators (2200 mm × 860 mm × 1880 mm) and had full access to feed and water.

The low pathogenicity avian influenza virus (LPAIV) H9N2 (A/Chicken/Shanghai/H514/2017) was used in the SHVRI laboratory (abbreviated as H514). It was isolated and stored by the Research Team at the SHVRI. For experimental use, the H514 was propagated in 10-day-old SPF embryonated chicken eggs (ECEs) (Beijing Merial Vital Laboratory Animal Technology Co., Ltd. Beijing, China). The modified viruses rH514-P29.1 and rH514-P29.2 were rescued in the SHVRI laboratory. Viral titers were calculated as median egg infectious doses (EID_50_).

### 2.2. The Preparation of the Recombinant Plasmids

We initially obtained three copies of P29 (Appendix A) with four GGGGS flexible linkers via gene synthesis via the GENEWIZ company (Suzhou, China). All linkers used in this study were GGGGS but coded by different nucleotides (Appendix A).

The synthesized P29.3 served as a PCR template. Copies 1, 2, and 3 of P29 were PCR-amplified using specific primers (Appendix A). Next, copies 1, 2 and 3 of P29 were inserted between the 3′ end of the HA signal peptide and the nucleotides encoding the N-terminal domain of the HA1 ectodomain of H514 HA (Figure 1A). The HA and recombinant HA-P29.N (N = 1, 2, 3) were cloned into pCAGGS plasmids using specific primers (Appendix A), named pCAGGS-HA, pCAGGS-HA-P29.1, pCAGGS-HA-P29.2, and pCAGGS-HA-P29.3, respectively. The recombinant plasmids were amplified in DH5α and extracted using a Plasmids Maxi Kit (QIAGEN, Hilden, Germany) according to the manufacturer’s instruction.

The eight gene segments of H514 were inserted into the vRNA-mRNA bidirectional transcription vector PHW2000. The gene segments of the recombinant HA-P29.N were inserted into the PHW2000 vector.

### 2.3. Western Blotting and Indirect Immunofluorescence Assay Analysis

We used western blotting (WB) and an indirect immunofluorescence assay (IFA) to identify the expression of the HA and HA-P29.N proteins. Leghorn male hepatoma (LMH) cells were seeded in 6-well plates (1 × 10^6^/mL/well) and transfected with 1 μg pCAGGS-HA-P29.N of plasmids or inoculated with 100 μL 10^6^EID_50_ of rescued modified virus. After 24 h, the cells were washed and treated with a SDS-PAGE loading buffer for WB or paraformaldehyde to stabilize the cells for the IFA.

To conduct WB, the harvested cells were treated with a SDS-PAGE loading buffer and subjected to SDS-PAGE. The separated proteins were electroblotted on polyvinylidene fluoride (PVDF) membranes and then blocked with 5% skimmed milk dissolved in 0.5% phosphate-buffered saline with Tween 20 (PBS-T). The membranes were probed with an anti-H514 HA monoclonal antibody (2F10) that was cloned and conserved in the SHVRI’s laboratory. After five washes using PBS-T, the membranes were then probed with anti-mouse IgG-HRPs (Sigma, St. Louis, MO, USA). The HA glycoprotein bands were visualized after the addition of ECL detection reagents. For the IFA, the transfected or inoculated cells were washed twice with phosphate-buffered saline (PBS). Paraformaldehyde (4%) was added to stabilize the cells and then incubated with 2F10 and later with fluorescence conjugated goat anti-mouse immunoglobulin G (Sigma, St. Louis, MO, USA). The results were observed by inverse microscopy (magnifications ×20). The densities were normalized to those of β-actin and calculated as HA-P29.N/HA expression ratios by using ImageJ (Version 1.53m).

### 2.4. Reverse Transcription qPCR Analysis

Reverse transcription qPCR (RT-qPCR) analysis was conducted to quantify the mRNA levels of chicken interferons (chIFNs). The total cDNA was generated from total mRNA extracted from transfected cells using the random 9 primer. Specific primers and probes were designed using the online tool (Appendix A). Chicken β-actin served as a reference gene. For each gene, the cycle threshold (Ct) values of different treatments at each time point were normalized to the respective endogenous control to get the ΔCt value. The difference in ΔCt value between the stimulated and control group was calculated (ΔΔCt). Quantification of mRNA levels from each resultant cDNA was expressed as fold changes (2^−ΔΔCt^) [42,43,44]. The RT-qPCR was 100% efficiency.

### 2.5. Virus Growth

The growth characteristics of the rH514, rH514-P29.1, and rH514-P29.2 in eggs were examined. The rH514, rH514-P29.1, or rH514-P29.2 were inoculated into SPF eggs that were between 9 to 11 days old at 10^4^ EID_50_. Allantoic fluid was harvested at 12, 24, 48, and 72 h-post inoculation (p.i). The harvested viruses’ titers were measured by calculating the TCID_50_ as previously described [45]. Briefly, a series of 10-fold dilutions of the samples were prepared in an EMEM medium with 1 mg/mL penicillin and 1 mg/mL streptomycin, and then inoculated into the MDCK cells (37 °C, 5% CO_2_). After 48 h of incubation, a HA assay using 0.5% chicken RBC in PBS was done to identify whether these samples contained the virus. The viral titers were calculated using the Reed–Muench method [46].

### 2.6. The Inactivated Vaccine Formation

The rH514 (EID_50_ = 9.50 Log_10_/mL), rH514-P29.1 (EID_50_ = 9.50 Log_10_/mL), and rH514-P29.2 (EID_50_ = 9.50 Log_10_/mL) were inactivated with 1:2000 β-propiolactone (BPL) by being shook constantly for 16 h at 4 °C. The residual β-propiolactone was evaporated at 37 °C for 2 h, and then three eggs were inoculated with 0.1 mL of the inactivated viruses. The eggs were then incubated for 48 h to confirm the loss of infectivity by an HA assay. The inactivated viruses were then mixed with a water-in-oil Montanide VG71 (0.85 g/cm^3^) adjuvant (SEPPIC, Brittany, France) at a volume ratio of 3:7 according to the manufacturer’s instructions [47].

### 2.7. Passively Transferred Antibody (PTA) Model and Animal Experiment

The PTA model was developed as previously described to mimic MDA in 1-day-old SPF chickens [29]. Hyperimmune sera containing H514-specific antibodies was generated by subcutaneous injections into 5-week-old SPF chickens with the inactivated vaccine made with H514 (0.5 mL/chicken) three times, with a 2-week interval. A total of 0.3 mL of the sera containing H514-specific antibodies (HI = 12 log_2_) was transferred intravenously into 1-day-old SPF chickens to achieve antibody titters of approximately 9 log_2_, which was similar to the high titters of natural MDA in 1-day-old commercial chickens detected in poultry. Negative sera that did not contain H514-specific antibodies were used as a negative control.

A group of 1-day-old chickens, with or without PTA (N = 6/group), were inoculated subcutaneously in the neck with 0.1 mL of the inactivated vaccines made with rH514, rH514-P29.1, or rH514-P29.2. Blood samples were then collected weekly and sera were separated for the detection of the chIFNs and rH514-specific antibodies. After 14 days of vaccination, peripheral blood mononuclear cells (PBMCs) were isolated and used for flow cytometry (FCM). Chickens with PTA were intranasally challenged with 10^6^ EID_50_ of H514 (0.1 mL/chicken) 28 days after they were vaccinated. Oronasal and cloaca swabs were collected at 3 and 5 days post-challenge (d.p.c). At the end of the experiments, all animals were euthanized.

### 2.8. Flow Cytometry

PBMCs were isolated by using a chicken peripheral blood lymphocyte isolation reagent kit (P8740, Solarbio, Beijing, China) according to the manufacturer’s instructions. A 15 μL volume of a panel of conjugated monoclonal antibodies, listed below against chicken cell surface markers, was added to 100 μL of PBMCs in a falcon tube and incubated in the dark for 15 min. All monoclonal antibodies were obtained from Invitrogen, USA. The panel included the following: anti-CD3-FITC (MA5-28696), anti-CD4-PE (MA5-28686), and anti-CD8-PE-Cyanine5 (MA5-28727). After being washed with PBS and centrifugation for 5 min at 300× *g*, the PBMCs were analyzed on an ACEA NovoCyte™ (BD Biosciences) flow cytometer. The raw data were gathered and analyzed using FlowJo10.10.0.

### 2.9. The ChIFNs ELISA Assay

The chIFNs (α, β, and γ) in the sera of vaccinated chickens were detected by using chIFN-α (SEKCN-0098), chIFN-β (SEKCN-0099), and chIFN-γ (SEKCN-0162) ELISA kits (Solarbio, Beijing, China), respectively, according to the manufacturer’s instructions.

### 2.10. The Hemagglutination Inhibition (HI) Assay

The antibodies were tested using the HI assay as previously described [48]. The BPL-inactivated H514 virus was used as a target antigen and diluted to standard HA units (8 HA in 50 μL). Serum samples were diluted in serial 2-fold dilutions and 0.5% chicken red blood was used in the HI assay.

### 2.11. Detection of Virus from Oronasal and Cloaca Swabs

To calculate viral shedding post-challenge, oronasal and cloaca swabs were collected at 3 and 5 d.p.c. Viral shedding was measured by calculating the TCID_50_ as mentioned above.

### 2.12. Statistical Analysis

Statistical analyses were performed using GraphPad Prism version 10.0 for Windows (GraphPad Software, San Diego, CA, USA). Significant differences were calculated using one-way ANOVA followed by a Tukey post-hoc test using SPSS software (Windows v16.0). All data follow the normal distribution and *p* ≤ 0.05 was considered to be significant.

## 3. Results

### 3.1. The Construction of the Recombinant Plasmids Expressing HAs Fused Different Copies of P29

To explore whether the P29-associated H9N2 AIV inactivated vaccine can overcome MDA interference in chickens, 1, 2 or 3 copies of P29 (Appendix A) were inserted into the HA gene segment, located behind the signal peptide (SP) of H9N2 AIV (A/Chicken/Shanghai/H514/2017, abbreviated H514). The structure is illustrated in Figure 1A. The gene segments of HA and HA-P29.N were cloned into the eukaryotic expression vector pCAGGS, named pCAGGS-HA and pCAGGS-HA-P29.N, respectively. The agarose and polyacrylamide gels demonstrated that the HA (1683 bp), HA-P29.1 (1800 bp), HA-P29.2 (1902 bp), and HA-P29.3 (2004 bp) gene segments were successfully integrated into the pCAGGS vector (Figure 1B). The sequence analysis revealed that there were no mutation and the entire recombinant sequences were fully consistent with our expectations.

The pCAGGS-HA and pCAGGS-HA-P29.N plasmids were transfected into LMH cells. The western blotting (WB) results indicated that the proteins HA, HA-P29.1, HA-P29.2, and HA-P29.3 were detected in the transfected cells. However, expression of the HA-P29.3 proteins was weak (Figure 1C). The indirect immunofluorescence assay (IFA) in transfected LMH cells also confirmed the expression of HA and HA-P29.N proteins. The IFA results showed that the HA-P29.2 proteins exhibited the highest fluorescence intensity, while the fluorescence intensity of the HA-P29.3 proteins was weak (Figure 1D).

**Figure 1 vaccines-13-00099-f001:**
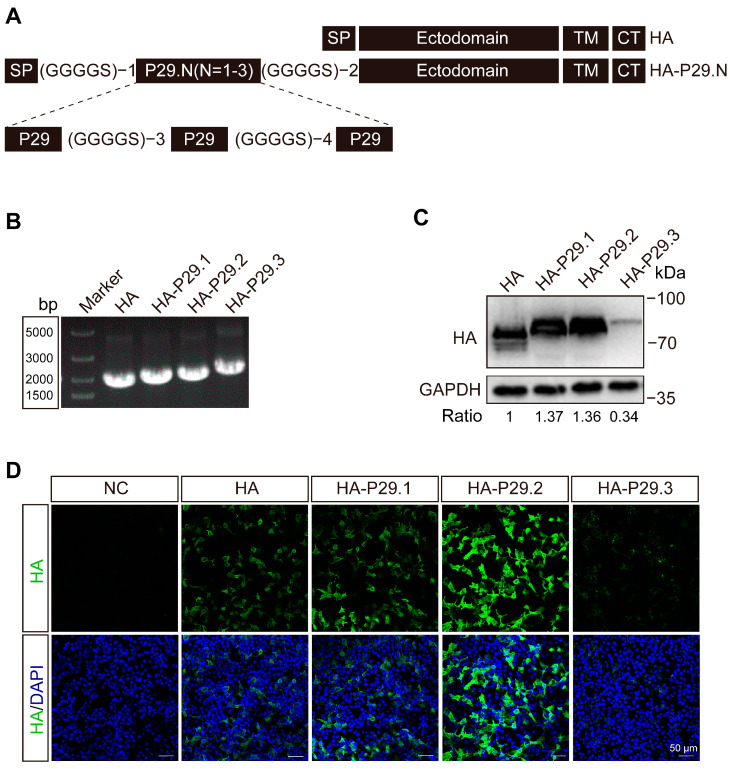
Development and identification of the recombinant HA-P29.N proteins. (**A**) The schematic of HA-P29.N gene segment. One, two, or three copies of P29 were inserted into the HA gene segment behind the signal peptide (SP). The flexible linker GGGGS, coded by different nucleotides, was used to connect P29 with the other part of HA. (**B**) The gel electrophoresis of the HA and HA-P29.N. The HA and HA-P29.N were PCR-amplified using the same primers but different plasmids as templates. (**C**) WB analysis of the recombinant HA-P29.N proteins. LMH cells were transfected with pCAGGS-HA, pCAGGS-HA-P29.1, pCAGGS-HA-P29.2, or pCAGGS-HA-P29.3 plasmids. After 24 h of transfection, cells were harvested to examine HA, HA-P29.1, HA-P29.2, and HA-P29.3 proteins by WB. (**D**) The IFA detection of the HA and HA-P29.N proteins. LMH cells were transfected with pCAGGS-HA, pCAGGS-HA-P29.1, pCAGGS-HA-P29.2, pCAGGS-HA-P29.3, or vector only as negative control. After 24 h of transfection, cells were harvested to examine HA and HA-P29.N proteins by IFA.

### 3.2. The HAs Fused Two Copies of P29 Promote the Expression of Type I chIFNs

IFNs are known to be positive stimulators of immune responses through their provision of co-stimulatory signals to lymphocytes that recognize their antigen via B cell receptors or T cell receptors engagement [34,49]. Therefore, RT-qPCR was used to quantify the mRNA expression of chIFNs in LMH cells transfected with pCAGGS-HA and pCAGGS-HA-P29.N plasmids. The chIFN-α mRNA expression stimulated by pCAGGS-HA-P29.2 was significantly higher than that stimulated by the other plasmid and was 100 times higher than that stimulated by vector only. The chIFN-β mRNA expression stimulated by both pCAGGS-HA-P29.2 and pCAGGS-HA-P29.3 was almost 300 times higher than that stimulated by vector only and was significantly higher than in cells transfected with pCAGGS-HA or pCAGGS-HA-P29.1. The chIFN-γ mRNA expression in cells transfected with pCAGGS-HA-P29.3 was significantly higher than in the other transfected groups, although the fold change was low (Figure 2).

### 3.3. The Generation of a Modified H9N2 Viruses Whose HA Fused Different Copies of P29

To develop a novel H9N2 AIV inactivated vaccine to overcome MDA interference, we cloned the recombinant HA-P29.N gene segment into the vRNA–mRNA bidirectional transcription vector PHW200. Using the backbone of H514 and the eight-plasmid system [50], we successfully rescued two modified H9N2 viruses, referred to as rH514-P29.1 and rH514-P29.2, respectively. We failed to rescue the modified rH514-P29.3. This is probably because of the low expression of HA-P29.3 proteins identified by WB (Figure 1C) and the IFA (Figure 1D). The parental H514 H9N2 virus was rescued as a control (abbreviated as rH514). The HA titers of the rH514-P29.1 and rH514-P29.2 was 2^10^ and 2^9^, respectively, which were similar to that of rH514 (2^10^). The three rH514, rH514-P29.1, and rH514-P29.2 viruses have the same EID_50_ titers (Appendix A). The results of the IFA (Figure 3A) and WB (Figure 3B) showed that the rH514-P29.1 and rH514-P29.2 express HA-P29.1 and HA-P29.2 proteins efficiently, respectively.

To determine whether the P29.N insertion affected the replication properties of these modified H9N2 viruses, we analyzed the growth kinetics of rH514, rH514-P29.1, and rH514-P29.2 in embryo eggs. Our results showed that the growth kinetics of rH514-P29.1 and rH514-P29.2 followed the same trend as those of rH514, with the highest viral titer being reached at 48 h post-inoculation (Figure 3C). Moreover, we assessed the genetic stability of rH514, rH514-P29.1 and rH514-P29.2 by continuously propagating them in 9 to 11 days old embryo eggs. The allantoic fluid from these virus-infected eggs were harvested at every passage up to 20 generations. The expression of HA, HA-P29.1, and HA-P29.2 proteins were identified by WB. The results showed that the HA, HA-P29.1, and HA-P29.2 proteins could be stably expressed and inherited in rH514, rH514-P29.1, and rH514-P29.2 recombination viruses, respectively (Figure 3B).

### 3.4. The rH514-P29.1 and rH514-P29.2 Inactivated Vaccines Promote the Secretion of Type I chIFNs

Since IFNs positively stimulate immune responses, the secretion of chIFN stimulated by the rH514, rH514-P29.1, or rH514-P29.2 inactivated vaccines were assessed in chickens with or without MDA. To ensure a uniform level of MDA in chickens, passively transferred rH514-specific antibody (PTA) were used to mimic MDA in 1-day-old SPF chickens as described previously [39,40,41]. The animal experiments were performed according to the strategies illustrated in Figure 4.

After 28 days of vaccination, serum was collected and the secretion of chIFNs in each group was measured using ELISA kits. The results indicated that, in chickens without MDA, the rH514-P29.1 or rH514-P29.2 inactivated vaccines stimulated significantly higher both chIFN-α and chIFN-β secretion than the vaccine made with rH514 (Figure 5A). Similarly, in chickens with PTA, the vaccine made with rH514-P29.2 stimulated significantly higher both chIFN-α and chIFN-β than the vaccines made with rH514 or rH514-P29.1 (Figure 5B). ChIFN-γ was not detected in any groups under any conditions.

### 3.5. The rH514-P29.2 Inactivated Vaccines Stimulates Robust Adaptive Immunity in Chickens with PTA

To evaluate whether the inactivated vaccines made with rH514-P29.N could induce strong cellular immune responses, blood samples were collected from each group and peripheral blood mononuclear cells (PBMC) were separated. Flow cytometry analysis was performed on PBMCs to evaluate the cellular immune responses in each group. The gating strategy for lymphocyte identification is illustrated in Figure 6.

To detect the T cells immune responses, the percentage of CD4+ and CD8+ T cells in PBMC was calculated based on these gating strategies by using the FlowJo 10.10.0. software. An overview of the proportion of CD4+ and CD8+ T cells in chickens with or without PTA are shown in Figure 7A and Figure 7B, respectively. Compared with the rH514 inactivated vaccine, the rH514-P29.1 and rH514-P29.2 inactivated vaccines induced significantly higher CD4+ T cell immune responses in chickens with PTA (Figure 7C), and stronger CD8+ T cells immune responses in chicken both with and without PTA (Figure 7D).

To assess the effectiveness of the rH514-P29.1 and rH514-P29.2 inactivated vaccines in inducing humoral immune responses, we collected serum samples from each group weekly after vaccination. The H514-specific antibodies in serum were measured using HI assay. The results showed that in chickens without PTA, the rH514-P29.1 and rH514-P29.2 inactivated vaccines induced antibodies more quickly and at higher levels than the rH514 inactivated vaccines after 21 days of vaccination (Figure 8A). In chickens with PTA, only the rH514-P29.2 inactivated vaccine induced significantly higher antibody levels than the rH514 inactivated vaccine after 21 days of vaccination (Figure 8B).

### 3.6. The rH514-P29.2 Inactivated Vaccine Reduces Viral Shedding in Chickens with PTA After Challenge

To gain insight into overcoming MDA interference, vaccinated chickens with PTA were challenged with rH514 after 28 days of vaccination. Following this challenge, the oronasal and cloaca swabs were collected at 3 and 5 d.p.c to assess viral shedding. The results showed that the rH514-P29.2 inactivated vaccine significantly reduced viral shedding at 3 d.p.c (Figure 8C). Viral shedding was not detected in the cloaca swabs and any groups at 5 d.p.c.

## 4. Discussion

An antigen bearing two and three copies of C3d increase the immunogenicity of the antigen by 1000 and 10,000 times, respectively [27]. The minimum-binding domain of C3d, P28 in mammals and P29 in avian species, can also significantly increase the immunogenicity of antigens [25,30,31]. In the present study, we found that HA proteins fused one (HA-P29.1) and two (HA-P29.2) copies of P29 stimulated strong secretion of type I chIFNs in vitro. Consequently, we rescued two modified H9N2 AIVs based on the H514 strain, which express HA-P29.1 (rH514-P29.1) and HA-P29.2 (rH514-P29.2) proteins efficiently. To evaluate whether the inactivated vaccines made with rH514-P29.1 or rH514-P29.2 could overcome MDA interference, the inactivated vaccines were used to immunize 1-day-old chickens with PTA. The results showed that the rH514-P29.2 inactivated vaccines induced strong type I chIFNs expression and robust adaptive immune responses in chickens with and without PTA. More importantly, the rH514-P29.2 inactivated vaccine significantly reduced viral shedding compared with the vaccine without P29 in chickens with PTA, which suggests that vaccine antigens fused with two copies of P29 can overcome MDA interference in chickens.

MDA is one of the reasons for the failure of vaccination in the field [12,21,51]. Considerable research has been conducted to develop new vaccines that can overcome MDA interference. The CpG ODN (TLR-21 agonist) is reported to overcome MDA interference when used as an adjuvant for the H9N2 inactivated vaccine in chickens [52]. Besides TLR agonists, new vaccines conjugating antigens to single chain fragment variable antibodies against chicken antigen presenting cell receptor CD83 can also overcome MDA interference in chickens [53]. Due to the characters of cell-associated nature and the nature of replication, turkey herpesvirus (HVT) is used as a live vaccine vector to bypass MDA interference in chickens [40,41,54,55]. However, no commercial vaccines that can overcome MDA interference are extensively applied in the field. This is probably because of the high cost and safety issues associated with these vaccine candidates, which need frequent purification or cold chain preservation. Traditional inactivated vaccines are low-cost but are sensitive to MDA. Therefore, in the present study, we designed a novel inactivated vaccine that stimulated robust immune responses in the presence of MDA by increasing the immunogenicity of HA proteins.

The immunogenicity of antigens is crucial for vaccines to induce strong immune responses. To enhance the immunogenicity of HA proteins, we fused different copies of P29 behind the signal peptide (SP) of HA proteins. We found that 1 (HA-P29.1) and 2 (HA-P29.2) copies of the P29 fusion increased the immunogenicity of the recombinant HA proteins by inducing type I chIFN mRNA expression in LMH cells. However, 3 copies of P29 fused HA proteins (HA-P29.3) were hardly expressed in vitro. This indicates that the insertion of 3 copies of P29 (107 amino acids) into HA proteins may have a big influence on the structure of HA proteins. The whole gene fragment of HA is too small to fit in many extra polypeptide segments, even if this is behind the SP where there is the most flexibility [56]. As a consequence, the modified H9N2 AIV expressing HA-P29.3 is not successfully rescued by reverse genetics. However, in contrast to our study, other researchers found that the H1N1 AIV HA proteins fused G proteins of central conserved-domains (114 amino acids) of the respiratory syncytial virus (RSV) behind the SP induce dominant antibodies and protect animals from RSV infection [57]. It is probable that the HA proteins of the H9N2 AIV may be less flexible than those of the H1N1 AIV, which are compatible with many more amino acids. Beside the position behind SP, the head-domain of HA proteins is also a place to insert foreign antigens. Li et al. reported that inserting 12 amino acids into the head-domain of the H1N1 AIV HA proteins induces strong B- and T-cell immune responses [58]. However, we found that the insertion of partial P29 (15 or 14 amino acids) into the head-domain will abolish the immunogenicity of the H9N2 AIV HA proteins. This may be because the amino acids we inserted destroyed the structure of the HA protein. Overall, the position behind the SP is more flexible than the head-domain of HA proteins.

The vaccine made with rH514-P29.1 or rH514-P29.2 significantly increased the expression of type I chIFN and stimulated strong adaptive immune responses in chickens with and without PTA. The results from our study are in line with previous studies using different methods in other species. The administration of an mRNA vaccine encoding spike proteins of SARS-CoV-2 attached with 3 copies of C3d induces a 10-times higher level of antibodies than the same mRNA vaccine without C3d in mice [26]. Furthermore, a polypeptide linear epitope G5 fused the molecular adjuvant P28 (P29 in chickens) enhanced virus neutralizing antibodies and induced a specific T-cell proliferation response [31]. Similarly, F proteins of the Newcastle disease virus were fused with different copies of P29 (N = 1–6) promoting the secretion of antigen-specific antibodies, which protected chickens from infection [25].

Until now, the immunogenicity of antigens fused different copies of C3d or P28/P29 has only been explored in animals without MDA. Little is known about their immunogenicity in animals with MDA. Lee et al. once inserted part P28 into VP1 of foot-and-mouth disease to develop a new inactivated vaccine. The inactivated vaccine induced robust adaptive immune responses in pigs with MDA by a booster-vaccination [33]. By using PTA to mimic MDA, we found that the inactivated vaccine made with rH514-P29.2 induced significantly strong adaptive immune responses, reducing viral shedding in chickens with PTA, which suggests that the vaccine antigens fused two copies of P29 can decrease MDA interference with only one-shot of vaccination. Nevertheless, more experimentation is needed to explore the efficacy of these vaccines in the presence of MDA in the field. One limitation that should be mentioned is that we did not figure out why the viral shedding was not be detected in any group at 5 d.p.c. Another limitation is that it is still unknown why the expression of HA-P29.3 were extremely low. Using computational or experimental structural analysis may help us to understand this phenomenon in the future.

In conclusion, this study demonstrated that the rH514 HA proteins fused two copies of P29 stimulated high expression of type I chIFN in vitro and in vivo. The inactivated vaccine made with rH514-P29.2 induced robust adaptive immune responses and reduced viral shedding in chickens in the presence of PTA. We firstly evaluated the immunogenicity of antigens fused different copies of P29 to overcome MDA interference. Most importantly, the novel vaccine is easy to proliferate, develop, transport and perform, which can reduce labor cost and money. Therefore, it is worth further evaluating the efficacy of other subtypes of AIVs fused with different copies of P28/P29 in different species to overcome MDA interference in the future.

## Figures and Tables

**Figure 2 vaccines-13-00099-f002:**
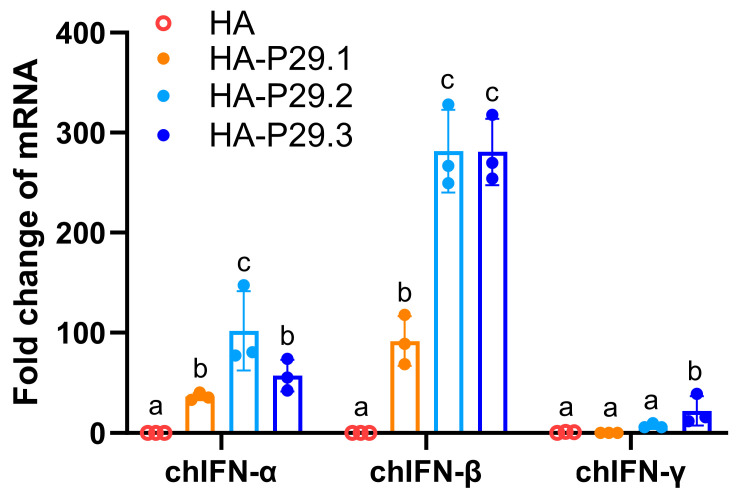
The mRNA expression of chIFN-α, β and γ stimulated by HA and HA-P29.N proteins in transfected LMH cells. LMH cells were transfected with pCAGGS-HA, pCAGGS-HA-P29.1, pCAGGS-HA-P29.2, pCAGGS-HA-P29.3, or a vector only as negative control. After 12 h of transfection, cells were harvested to examine the relative mRNA expression of chIFN-α, β and γ by RT-qPCR. Different letters denote significant differences among each group.

**Figure 3 vaccines-13-00099-f003:**
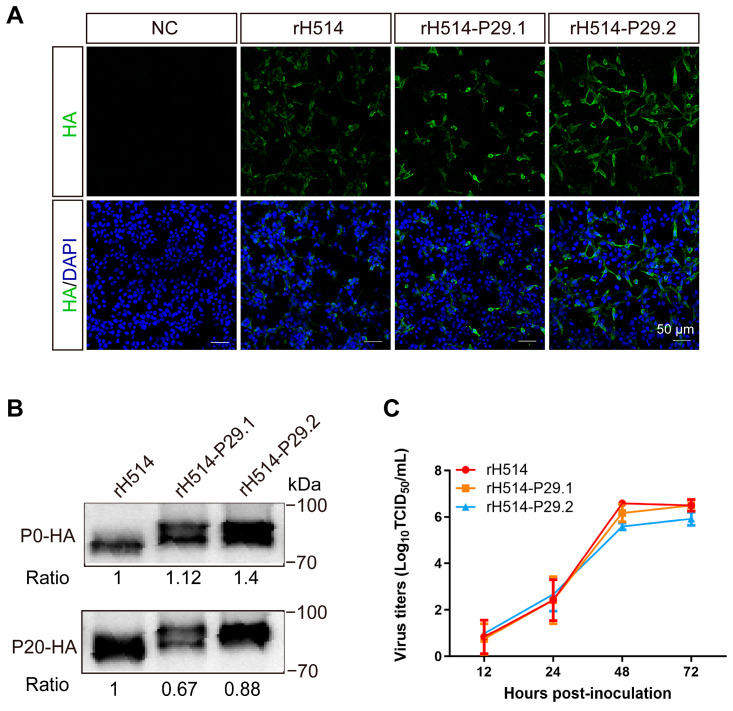
Identification and characterization of the modified rH514-P29.N and parental rH514 viruses in vitro. (**A**) The IFA detection of the rH514 and rH514-P29.N recombinant viruses. LMH cells were inoculated with the rH514, rH514-P29.1, rH514-P29.2, or PBS served as a negative control. Cells were harvested to examine the HA and HA-P29.N proteins by an IFA after 24 h of inoculation. (**B**) WB analysis of the rH514 and rH514-P29.N recombinant viruses. The viruses were continuously propagated in 9 to 11-day-old eggs up to 20 passages (P20). The allantoic fluid from P0 and P20 virus-infected eggs were harvested and subjected to WB to examine HA and HA-P29.N proteins. (**C**) Growth curve of the rH514 and rH514-P29.N recombinant viruses. The 9 to 11 cc old eggs were inoculated with 10^4^ EID_50_ of the rH514 or rH514-P29.N. The allantoic fluid from infected eggs were harvested at 12, 24, 48, and 72 h.p.i. and titrated in MDCK cells.

**Figure 4 vaccines-13-00099-f004:**
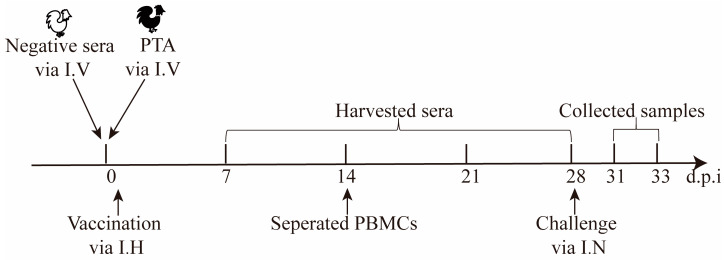
The schematic of animal experimental strategy. Passively transferred 0.3 mL of H514-specific antibody (PTA) into 1-day-old chickens (N = 6/group) via intravenous injection to mimic MDA. Chickens were immediately and subcutaneously vaccinated with 0.1 mL of the rH514, rH514-P29.1, or rH514-P29.2 inactivated vaccines. Sera and PBMC were collected at indicated time points. Chickens were intranasally challenged with 10^6^EID_50_ of H514 (0.1 mL/chicken) at 28 d.p.i. Oronasal and cloaca swabs were collected at indicated time points.

**Figure 5 vaccines-13-00099-f005:**
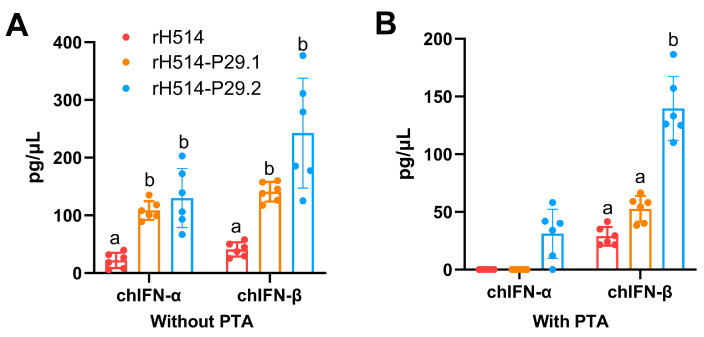
The chIFN-α and β expression in sera of vaccinated chickens with and without PTA. The chickens without (**A**) and with (**B**) PTA were inoculated with rH514, rH514-P29.1, or rH514-P29.2 inactivated vaccines. The amount of chIFN-α and -β in sera collected at 28 d.p.i were detected using ELISA kits. Different letters denote significant differences among each group.

**Figure 6 vaccines-13-00099-f006:**
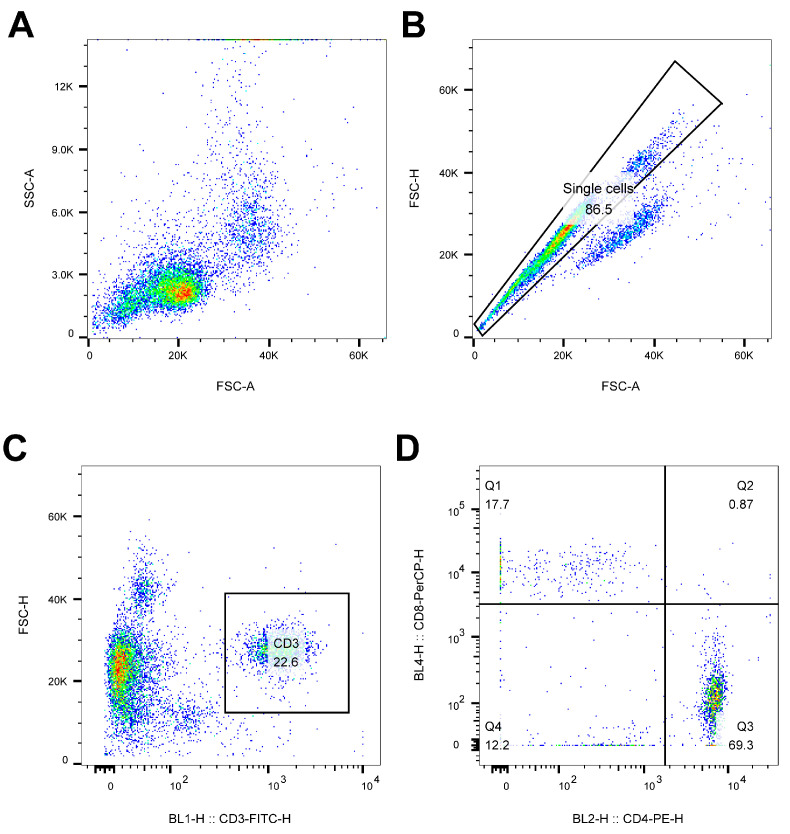
Gating strategy for lymphocytes identification by flow cytometry analysis. Putative lymphocytes were gated based on the light scatter properties (**A**) and doublet cells were excluded based on FSC-A versus FSC-H (**B**). T cells were identified as being CD3+ (**C**) and two subsets were identified in this way: CD3+CD4+CD8- and CD3+CD4-CD8+ T cells (**D**).

**Figure 7 vaccines-13-00099-f007:**
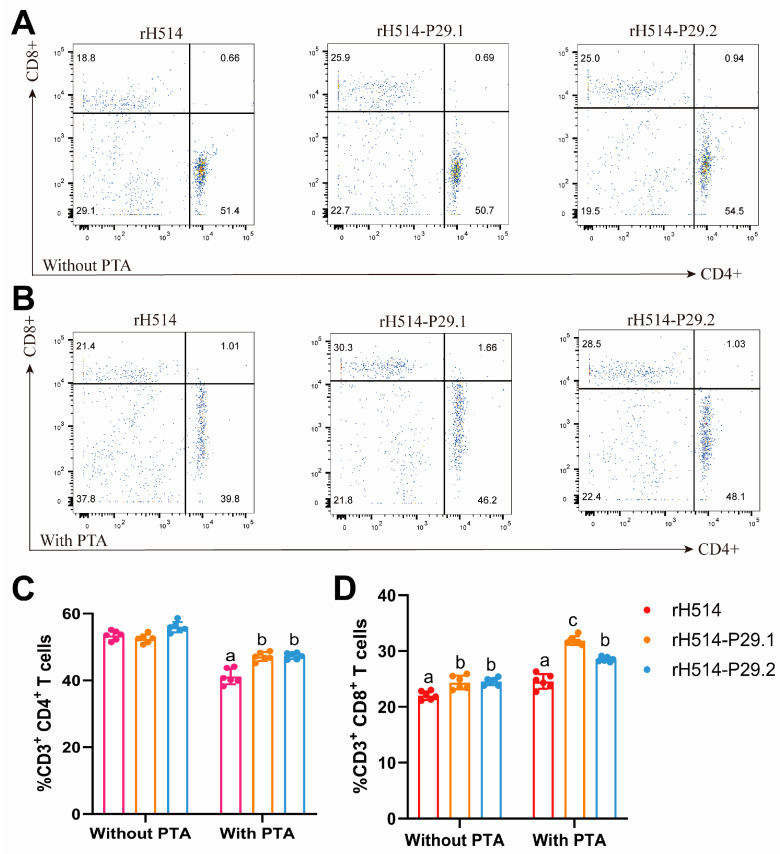
The proportion of CD4+ and CD8+ in PBMCs of vaccinated chickens with and without PTA. The chickens with and without PTA were inoculated with rH514, rH514-P29.1, or rH514-P29.2 inactivated vaccines. PBMCs were collected at 14 d.p.i. and subject to FCM to show an overview of CD4+ (**A**) and CD8+ (**B**) T cells in each group. The total proportions of CD4+ (**C**) and CD8+ (**D**) T cells were calculated and presented. Different letters denote significant differences among each group.

**Figure 8 vaccines-13-00099-f008:**
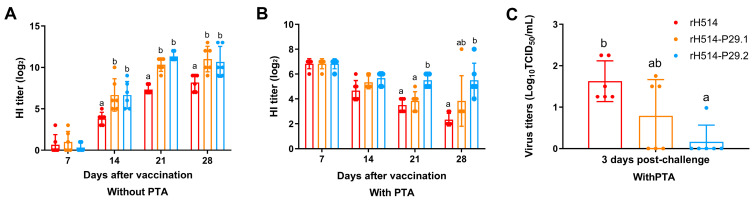
The humoral immune responses in vaccinated chickens with and without PTA and viral shedding after challenge. Chickens were inoculated with the rH514, rH514-P29.1, or rH514-P29.2 inactivated vaccines and sera were collected weekly. The H514-specific antibodies in chickens without (**A**) and with (**B**) PTA were evaluated by HI. (**C**) The viral titers from oropharyngeal swabs of vaccinated and challenged chickens with PTA were detected at 3 d.p.c. Different letters denote significant differences among each group.

## Data Availability

All materials are available upon request to interested researchers.

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
