# Peer review of "The Generation of a H9N2 Avian Influenza Virus with HA and C3d-P29 Protein Fusions and Vaccine Development Applications"

_vaccines, 2025, doi:10.3390/vaccines13020099_

Round 1

Reviewer 1 Report

Comments and Suggestions for Authors

The authors present a report on the immunogenicity of HA H9 constructs with 1-3 copies of the P29 unit from C3d in chickens. They demonstrate through in vitro and in vivo data that HA H9 with 2 P29 copies exhibits higher immunogenicity. This phenomenon is also observed in immunization experiments of chicken chicks after passive transfer of H9-specific antibodies. The authors hypothesize that enhanced B cell activation is attributable to the binding of the P29 units to CD21 on B cells.

The interference of maternal immunity with inactivated vaccines has significant negative effects on the use of vaccinations against HPAIV in poultry, underscoring the need for approaches to improve the efficacy of vaccinations. In the present manuscript, the authors transfer a system previously studied in virus infections in mammals to poultry, documenting comparable effects in a broad spectrum of investigation on humoral and cellular immunity. In a challenge experiment, the researchers also demonstrated significantly improved inhibition of virus excretion in vaccinated animals. The manuscript is suitable for publication; however, a thorough linguistic revision should be carried out to correct some language issues.

Examples of linguistic inaccuracies are

L 32 - Do the authors mean experimental vaccination challenge tests compared to field use of vaccines?

L364 – I do not understand the meaning of this sentence.

Another point of criticism is the differentiation between PTA and MDA. In the course of the manuscript, the authors increasingly treat this as identical, but there are still significant differences (e.g. Abstract L 22/23). In this respect, PTA should be used strictly. Whether the results can be repeated in MDA-positive chicks depends on corresponding follow-up experiments.

The P29.2 HA is obviously expressed at a higher ratio compared to the other constructs. Could the effects observed in LMH cells be due to the increased expression level as such?

Comments on the Quality of English Language

See above.

Reviewer 2 Report

Comments and Suggestions for Authors

Please take a look at the comments I've attached.

Reviewer 3 Report

Comments and Suggestions for Authors

The manuscript "Generation of a H9N2 avian influenza virus whose HA proteins fused C3d-P29 and its application in vaccine development" by Pan et al. used P29 domain of C3d to insert into the HA gene of the H9N2 virus and to use it as an inactivated vaccine. Authors have shown that 3 copies of P29 insertion in the HA gene induced higher immunogenicity and lower shedding of challenge virus in vaccinated chickens.

Although the strategies to use hyperimmune serum to mimic MDA in day-old SPF chicks have the advantage that a homogenous antibody level will be maintained among the birds in the group, in contrast, administrating hyperimmune serum by i/v to day-old chicks is an arduous task. Although the result looks promising in SPF birds, it will be interesting to see how the vaccine performed in commercial chickens having MDA.

I have the following minor comments:

  1. Please provide the background of using the H9N2 vaccine in breeder chickens in China and the rationale for using H9N2 as a vaccine candidate.
  2. MDA is one of the important factors in vaccine failure for IBD, in addition to AIV and NDV (Line 40). Provide some reference for IBD.
  3. Table 2 can be added as a supplementary table to save space.
  4. Line 105: Eghorn male hepatoma (LMH) should be Leghorn male hepatoma.
  5. Line 137: The harvested viruses were measured……… It should be the harvested virus titer.
  6. Line 259: rH514 (210) , HI titter should be 210
  7. In the discussion, lines 379 to 395, the MDA section could be shortened.
  8. In the future, authors should try to use commercial chickens having MDA.

Reviewer 4 Report

Comments and Suggestions for Authors

In this manuscript, the H9N2 subtype avian influenza virus vaccines with a C3d-P29 fuse HA protein were developed to overcome the maternal-derived antibody (MDA) interference in chickens. The results showed that rH514-P29.1 and rH514-P29.2 inactivated vaccine elicited significantly higher HAI titers than wild type rH514 inactivated vaccine. Animals in rH514-P29.2 inactivated vaccine group had significantly less viral shedding in the oronasal and cloaca swab samples than the animals in wild type rH514 inactivated vaccine group on day 3 post-challenge. However, more data is needed to support the conclusion proposed in the manuscript.

Major comments:

1.     Language and writing: Highly recommend to have a native speaker to help with the language. The logistics of writing also needs to be improved.

2.     Figure 8: The mock vaccine control group is missing especially for MDAs animal model. 

Minor comments:

3.     Line 13: “different”. Maybe “one or multiple” is better.

4.     Line 16: “pure”. Should be wild type.

5.     Line 19: “adaptive immune responses”. Should be “HAI antibody responses” as this is what you used in your study.

6.     Line 20: “with rH514 in chickens without MDAs”. “Naive chickens”?

7.     Line 21: “vaccine without P29”. “wild type vaccine”?

8.     Line 20-23 can break into 3 sentences.

9.     Line 98-102: No result in the M&M section.

10.  Line 108: How were the cells harvested for each assay?

11.  Line 114-115: “and then anti-mouse IgG-HRP (Sigma, United States).” Make this another sentence.

12.  Line 144: How much antigen were used? How to determine?

13.  Line 237: “empty plasmids”. Should be “vector only”. There is no vector only group in figure 2.

14.   Figure 2, 5, 7, 8: “P ≤ 0.05 was considered to be significant.” Don’t see stats in the figures.

15.  Figure 3B: low resolution

16.  Line 302: Different methods (qRT-PCR and ELISA) were used for determining chIFNs. Should keep consistence through the manuscript.

17.  Figure 8A: I understand this should be a naive model; it will be interesting to challenge those animals so that we can see response of viral infection to vaccination in both models (naive and MDAs).

18.  Figure 8B: It will be better to have HAI titers on day 0 as a baseline.

19.  Figure 8C: Are there any symptoms during the viral infection?

20.  Line 417-432: This should be in the introduction section. Besides, it is necessary to introduce C3d and how it contributes to enhance the immune responses (humoral and cellular) with or without the presence of MDAs.

21.  Line 433-434: This can be in the last paragraph of the introduction section.

Comments on the Quality of English Language

Highly recommend to have a native speaker to help with the language.

Reviewer 5 Report

Comments and Suggestions for Authors

In the article "Generation of a H9N2 avian influenza virus whose HA proteins fused C3d-P29 and its application in vaccine development", the authors developed new avian vaccines based on P29 to overcome existing maternal-derived antibodies (MDAs).

In line 82, you provided information about the sequence of the entire C3d, not P29. Please provide information about P29 itself, particularly its length.

Author Response

Comments 1: In line 82, you provided information about the sequence of the entire C3d, not P29. Please provide information about P29 itself, particularly its length.

Response 1: Thank you for pointing this out. We have provided the information of P29 in the revised draft in the supplementary table 1.

Round 2

Reviewer 5 Report

Comments and Suggestions for Authors

-
